# Activation of PI3K/Akt Signaling Pathway in Rat Hypothalamus Induced by an Acute Oral Administration of D-Pinitol

**DOI:** 10.3390/nu13072268

**Published:** 2021-06-30

**Authors:** Dina Medina-Vera, Juan Antonio Navarro, Rubén Tovar, Cristina Rosell-Valle, Alfonso Gutiérrez-Adan, Juan Carlos Ledesma, Carlos Sanjuan, Francisco Javier Pavón, Elena Baixeras, Fernando Rodríguez de Fonseca, Juan Decara

**Affiliations:** 1Instituto de Investigación Biomédica de Málaga (IBIMA), Hospital Universitario Regional de Málaga, UGC Salud Mental, Avda. Carlos Haya 82, Pabellón de Gobierno, 29010 Málaga, Spain; dina.medina@ibima.eu (D.M.-V.); juan_naga@hotmail.es (J.A.N.); rubentovar7@hotmail.com (R.T.); cristina.rosell@ibima.eu (C.R.-V.); juan.ledesma@uv.es (J.C.L.); javier.pavon@ibima.eu (F.J.P.); 2Facultad de Ciencias, Universidad de Málaga, 29010 Málaga, Spain; 3Facultad de Medicina, Universidad de Málaga, 29010 Málaga, Spain; 4Centro de Investigación Biomédica en Red de Enfermedades Cardiovasculares (CIBERCV), UGC del Corazón, Instituto de Investigación Biomédica de Málaga (IBIMA), Hospital Universitario Virgen de la Victoria, Universidad de Málaga, 29010 Málaga, Spain; 5Departamento de Reproducción Animal, Instituto Nacional de Investigación y Tecnología Agraria y Alimentaria, 28040 Madrid, Spain; agutierr@inia.es; 6Euronutra S.L. Calle Johannes Kepler, 3, 29590 Málaga, Spain; euronutra@euronutra.eu; 7Departamento de Bioquímica y Biología Molecular, Facultad de Medicina, Universidad de Málaga, 29010 Málaga, Spain; ebaixeras@uma.es

**Keywords:** inositol, D-Pinitol, insulin resistance, hypothalamus, PI3K/Akt pathway

## Abstract

D-Pinitol (DPIN) is a natural occurring inositol capable of activating the insulin pathway in peripheral tissues, whereas this has not been thoroughly studied in the central nervous system. The present study assessed the potential regulatory effects of DPIN on the hypothalamic insulin signaling pathway. To this end we investigated the Phosphatidylinositol-3-kinase (PI3K)/Protein Kinase B (Akt) signaling cascade in a rat model following oral administration of DPIN. The PI3K/Akt-associated proteins were quantified by Western blot in terms of phosphorylation and total expression. Results indicate that the acute administration of DPIN induced time-dependent phosphorylation of PI3K/Akt and its related substrates within the hypothalamus, indicating an activation of the insulin signaling pathway. This profile is consistent with DPIN as an insulin sensitizer since we also found a decrease in the circulating concentration of this hormone. Overall, the present study shows the pharmacological action of DPIN in the hypothalamus through the PI3K/Akt pathway when giving in fasted animals. These findings suggest that DPIN might be a candidate to treat brain insulin-resistance associated disorders by activating insulin response beyond the insulin receptor.

## 1. Introduction

Inositols (1,2,3,4,5,6-cyclohexanehexol) can be differentiated depending on six hydroxyl groups configuration being D-Pinitol (DPIN) a methylated at the 3-position OH [1]. DPIN is existing by nature plant inositol derivates being a bioactive compound with insulin-like effect [2]. DPIN is present in pine trees, legumes, seeds, and flowers having one of the highest antioxidant activities of the inositols [3]. Inositols, including DPIN, can be also found in mammalian as inositol glycans (IGs) [4], carbohydrates derived from glycolipids produced by insulin-sensitive cells in response to insulin treatment [5]. IGs were first identified in 1986 from bovine liver treated with insulin [4] when the analogous activity between IGs and insulin suggested that IGs may be involved in the transduction of insulin signals acting as second messengers [6]. Besides, IGs performs some insulin-like activities, including stimulation of lipogenesis, glucose transport, and glycogen synthesis [7,8]. It has been proposed that non-canonical insulin signaling might be mediated by the actions of these IGs, offering new alternatives for the treatment of diseases linked to carbohydrate metabolisms such as insulin resistance and Type 2 Diabetes [9]. The main and most studied inositols present in these glycans are myoinositol and D-chiro-inositol, being DPIN the 3-*O*-methyl form of DCI (Figure 1A). Additional research suggests that inositols might exert direct actions as insulin sensitizers [10]. Although most studies have been performed in peripheral tissues, the direct actions of these inositols, specially DPIN, in the brain are less understood.

Among plant-derived inositols, DPIN is a very interesting molecule. It is present in carob fruits (*Ceratonia siliqua*) in large concentrations [11], which allows its isolation and industrial production. Initial studies suggested that it is a bioactive compound with a positive effect on diabetic symptoms [12,13], in addition of having other biological activities such as antihyperlipidemic [14], anti-asthmatic [15], and antioxidant [16,17], among others. Since the effects of DPIN on central insulin signaling has not been addressed, we decided to analyze the effects of an acute oral DPIN administration on the insulin signaling cascade. If proven, it might open new alternatives for the treatment of chronic disorders where central insulin resistance plays a relevant role, such as Type 2 Diabetes and Alzheimer’s disease since they are associated with insulin resistance [18,19,20,21].

Hypothalamus has been selected in this study for being one of the main structures in the central nervous system involved in the control of glucose homeostasis and systemic energy balance. The important nuclei for the action of insulin within the hypothalamus are the arcuate nucleus and the paraventricular nucleus [22,23,24], expressing a high quantity of insulin receptors [25]. Apart from insulin, other metabolic hormones such as ghrelin [26] and leptin [27] has been identified by activating and interacting with intracellular signaling pathways in the hypothalamic neurons. In this sense, insulin, ghrelin, and leptin play an important role in the regulation of food intake and energy expense.

To evaluate the insulin-mimetic actions of DPIN, we focus on the Phosphatidylinositol-3-kinase (PI3K)/Protein Kinase B (Akt) signaling pathway (Figure 1B), which is part of the insulin cascade via Insulin Receptor Substrate 1 (IRS-1). Physiological growth factors, such as insulin, bind to its tyrosine kinase-like cell membrane receptor. A receptor-insulin complex is formed and this causes a conformational change in the receptor itself. The two intracellular domains phosphorylate each other by modifying their kinase activity and, therefore, the affinity of the two domains to other substrates. The adapter IRS-1 bind now the phosphorylated receptor. The p85 regulatory domain of the PI3K (p85-PI3K) binds the phosphorylated domains of IRS-1, activating PI3K, which is close to its membrane substrate Phosphatidylinositol-4,5-diphosphate (PIP2). PI3K now phosphorylates PIP2 generating phosphatidylinositol-3,4,5-triphosphate (PIP3) and inducing the recruitment and activation of Phosphoinositide-dependent kinase-1 (PDK1). PDK1 then phosphorylates Akt directly at the threonine 308 residue (T308), or indirectly at the serine 473 by mammalian Target of Rapamycin protein (mTOR) Complex 2 (mTORC2) to activate Akt. Akt phosphorylates Glycogen synthase kinase 3β (GSK-3β) to inactivate GSK-3β and therefore to activate glycogen synthase (GS), leading to glycogen synthesis. Akt also phosphorylates mammalian Target of Rapamycin Complex 1 (mTORC1) to activate it and to increase protein synthesis. Additionally, mTORC1 inhibits autophagy by several mechanisms. Besides, Akt phosphorylates other physiological substrates, which promotes cell survival and proliferation. The Akt pathway can be turned off at the beginning of the pathway through the phosphatase and Tensin homolog (PTEN) which downregulates PIP3. Another phosphatase that can turn off the pathway is the Protein phosphatase 2C (PP2C) which dephosphorylates Akt at threonine 308. By last, mTORC1 can also inhibit IRS-1 by phosphorylating IRS-1 at serine residues. This would represent a feedback inhibition of insulin signaling. Phosphorylation of Adenosine monophosphate-activated protein kinase-Thr172 (AMPK-Thr172) is required for the activity of AMPK. AMPK activity can be attenuated by phosphorylation of AMPK-Ser485/491 in response to insulin (acting through Akt) or in response to Cyclic adenosine monophosphate (cAMP) elevating agents that drive the activation of protein kinase A (PKA). Proteins belonging to PI3K/Akt pathway were evaluated in the present study under the hypothesis of DPIN acting intracellularly bypassing the IRS-1. All this cascade is being evaluated in the hypothalamus, an insulin-sensitive brain area with metabolic homeostasis regulation, acting as an integrative interface between peripheral organs and central nervous system processing.

In the present study, we explore the PI3K/Akt signaling pathway in the hypothalamus of Wistar rats after an acute oral administration of the natural inositol DPIN, considering the time dependence of the compound. Initially, we evaluated the concentration of DPIN, glucose and peripheral hormones in plasma. Second, we analyzed the phosphorylated and total forms of the signaling proteins. By last, we considered the metabolic sensors mTOR and Adenosine monophosphate-activated protein kinase (AMPK) in the hypothalamus and its possible regulation by DPIN post-administration. The outcome of this study assessed the potential regulatory effects of DPIN on the hypothalamic insulin signaling pathway and metabolic sensors, as a basis for establishing the potential utility of the inositol as a novel therapeutic target to treat brain insulin-resistance associated disorders.

## 2. Materials and Methods

### 2.1. Animals and Ethics Statement

All the experiments were performed on male Wistar rats (Crl:WI(Han)), weighing 350 ± 20 g and age of 4–5 weeks, obtained from Charles Rivers Laboratories (Barcelona, Spain). Animals were kept with free access to water and food under standardized environments for experimental rodent in the Animal Experimentation Service of the University of Malaga: 20 ± 2 °C temperature of room with 40 ± 5% relative humidity and light/dark cycle of 12 h with sunrise/sunset sequence. All the procedures detailed in this study were carried out strictly following the European Directive 2010/63/EU from the protection and care about used animals for scientific purposes from Spanish legislation (Real Decreto 53/2013) for the use of laboratory animals. The protocol was approved by the Research and approved by Ethics Committee for Animal Experiments of the University of Malaga (CEUMA), Spain. All of it in accordance with the ARRIVE guidelines [28], doing all possible efforts to minimize animal suffering and reducing the number of rats used per experimental group.

### 2.2. Preparation and Administration of D-Pinitol

DPIN (3-*O*-methyl-d-chiro-inositol, 98% purity) crystalline fine powder (Caromax^®^-D-Pinitol), was provided by Euronutra SL (https://www.euronutra.com/, accessed on 23 June 2021, Malaga, Spain). DPIN was dissolved in sterile water at a dose of 500 mg per Kg body weight (BW), to be administrated in a final volume of 1 mL/Kg by gavage. After 18 h of fasting, DPIN was administrated orally and the rats were sacrificed in groups (*n* = 8) at different times: 0, 60, 120 and 240 min (min) after DPIN administration. The control group (0 min) was administered only with water.

### 2.3. Samples Collection

Finalized the DPIN administration, all animals were anaesthetized with intraperitoneal administration (ip) of sodium pentobarbital in a dose of 50 mg/Kg body weight and finally sacrificed by decapitation. Blood was centrifuged (2100× *g* for 8 min, 4 °C) and plasma and brain samples were immediately frozen at −80 °C for late analysis [29].

### 2.4. Plasma DPIN Concentration

Plasma DPIN concentrations were performed and supervised by Medina Foundation Research Center (Granada, Spain) by means of specific method of liquid chromatography-mass spectrometry. Multiple Reaction Monitoring (MRM) mode with electrospray positive ionization was used to detect both analytes and internal standards. The limits of detection were 333 to 20,000 ng/mL of DPIN. The PK Solver 2.0 program was used to calculate the concentration of DPIN at each time point using a non-compartmental analysis of plasma data following extravascular input.

### 2.5. Protein Extraction

Medial basal hypothalamus was dissected immediately after sacrifice and frozen until analysis. Frozen samples (17 mg average weight) were homogenized in 1 mL of 4 °C RIPA lysis buffer (150 mM NaCl, 50 mM Tris-HCl, 1 mM EDTA, 1 mM Na_3_VO_4_, 1 mM NaF, 1% Triton, 0.5% NaDOC, 0.1% SDS) with a cOmplete™ Protease Inhibitor Cocktail (Roche, cat. Number: 11836145001) complement. The mixture was incubated at 4 °C for 2 h and centrifuged at 12,000 rpm at 4 °C during 15 min. Then, supernatant obtained was removed to a new clean tube for Bradford colorimetric method to determine the total protein concentration. Finally, each sample protein extracts were diluted 1:1 in loading buffer with DTT 2X (Bio-Rad, cat. Number: 1610737) heating at 99 °C for 5 min to later before perform electrophoresis.

### 2.6. Western Blot Analysis

We performed Western blot technique on hypothalamic tissue samples to quantify levels of the interested proteins of the insulin signaling pathway. Results are normalized with γ-Adaptin, a highly conserved protein and it is present in similar amounts in the four groups used as load control. For electrophoresis, tissue protein extracts (10–15 μg) were loaded on 4–12% Criterion™ XT Bis-Tris Precast Gels (Bio-Rad, cat. number: 3450124) at 80 V for 30 min and at 150 V during 2 h. Once the electrophoresis finished, the proteins were transferred onto 0.2 µm pore–size nitrocellulose membrane (Bio-Rad Bio-Rad, cat. number: 1620112) during 1 h at 80 V by wet transfer equipment systems (Bio-Rad). Then, membrane was carefully washed twice at least for 5 min in Tris-buffered saline with Tween (TBST) (150 mM NaCl, 10 mM Tris–HCl, 0.1 % Tween 20 and blocked with 2% Bovine Serum Albumin-TBST buffer (BSA-TBST) for 1h at room temperature on shaking platform. After that, the membrane was incubated overnight with the appropriate diluted in 2% BSA-TBST primary antibodies at 4 °C. Antibodies against IRS-1, phospho-IRS-1 (Ser307/612), PI3Kp85, GS, phospho-GS (Ser641), phospho-GSK-3β (Ser9), GSK-3β, PTEN, Akt, phospho-Akt (Ser473), AMPK-α, phospho-AMPK-α (Thr172), mTOR and phospho-mTOR (Ser2448) were purchased from Cell Signalling Technology; phospho-IRS-1 (Tyr896), phospho-PI3Kp85 (Tyr607), γ-Adaptin from Abcam; PKAα cat (C-20), phospho-PKAα (Thr198) from Santa Cruz Biotechnology, Inc.; PP2Cα from R&D Systems, Inc.; and, phospho-GSK-3β (Tyr279/Tyr216) from Merck Millipore (Appendix A for additional information regarding used antibodies). The next day, the membrane was washed thrice with TBST for 10 min. A suitable conjugated to horseradish peroxidase (HRP)/mouse secondary antibody (Promega) was diluted 1:10,000 in 2% BSA-TBST and promptly incubated with the membrane with gently shaking at room temperature for 1 h. To conclude, the membrane was washed again as indicated above and finally revealed with chemiluminescent reagent (Santa Cruz, Biotechnology Inc. cat. number sc-2048) during 5 min. Corresponding membrane-bound protein was visualized by Chemiluminescence method (ChemiDoc Imaging System, Bio-Rad). Each band was analyzed and quantified by densitometry using the image processing software ImajeJ (http://imagej.nih.gov/ij, accessed on 23 June 2021). Total protein signal levels were standardized to the signal level of each sample’s matching γ-Adaptin band on the same blot. The ratio of the signal produced with the phospho-specific antibody to the appropriate total protein antibody was used to determine the phosphorylation stage of protein.

### 2.7. Measurement of Biochemical Metabolites in Plasma

Biochemical glucose parameters were analyzed by biochemistry automatic analyzer (Hitachi Clinical Analyzer, Japan). The plasma levels of insulin and Insulin-like growth factor-1 (IGF-1) were determined with a rat enzyme-linked immunosorbent assay method (ELISA) using commercial kits (Millipore Corporation cat. number: EZRMI-13K and Invitrogen cat. number: ERIGF1). Glucagon levels in plasma were determined with a Rat Enzyme Immunoassay (EIA) kit (Sigma-Aldrich cat. number: RAB0202). All samples of serum were assayed in duplicate for each assay, and the results were reported in terms of a specific standard hormone. Homeostatic model assessment for Insulin Resistance (HOMA-IR) was calculated using the following formula: HOMA-IR= (fasting plasma insulin [uIU/mL] × fasting glycemia [mmol/L])/22.5) [30]. With the previously insulin and glucagon values obtained we calculated the insulin-glucagon ratio (IGR) [31].

### 2.8. Statistical Analyses

The results were expressed as mean ± standard error of the mean (SEM). For the statistical analysis of the different studies, a size of each group of 5–8 samples (*n* = 5–8) was taken into account. Statistical analysis was performed by the GraphPad Prism software, version 8 (GraphPad Software, Inc., San Diego, CA, USA). One-way Analysis Of Variance (ANOVA) was assessed for all results followed by Tukey’s *post hoc* multiple comparisons test. The *post hoc* tests only were conducted if F value in ANOVA reached a *p* less than 0.05 and homogeneity of variance was not statistically significant. The results were considered statistically significant at *p* < 0.05.

## 3. Results

### 3.1. Effects of Oral Administration of DPIN on Plasma Levels of Glucose, Insulin, Glucagon and IGF-1

First, we aimed to analyze the quantity of DPIN in plasma after its administration at 0, 60, 120 and 240 min and its effect on plasma levels of glucose, insulin, glucagon and IGF-1. The oral administration of 500 mg/kg DPIN resulted in a rapid rise of DPIN in plasma (Figure 2A) that peaked at 120 min (*p* < 0.001) after its administration, which is rapidly cleared from thereafter (*p* < 0.05). This rise of DPIN was associated with a decrease in the circulating concentrations of insulin (Figure 2B) (*p* < 0.001 and *p <* 0.01 for 60 and 120 min respectively) and maintaining stable glucose levels (Figure 2C). The glucagon plasmatic levels only show significant differences at 120 min after DPIN administration (*p* < 0.05) remaining unchanged in the first 60 min and returning to baseline at 240 min (Figure 2D). By last, IGF-1 levels in plasma do not show significant changes among the groups (Figure 2E).

### 3.2. Effects of Oral Administration of DPIN on Insulin Resistance Indexes

We next evaluated the Homeostatic model assessment for Insulin Resistance (HOMA-IR), a method used to quantify insulin resistance. As shown in Figure 3A, there is a remarkable decrease in the HOMA-IR ratio at 60 min (*p* < 0.001), being less significant compared with 120 and 240 min. Due to the opposing actions of glucagon and insulin upon hepatic glucose balance, also insulin-glucagon ratio (IGR) was calculated. The IGR ratio shows similar performance that HOMA-IR, a significative decrease after oral administration of DPIN (*p* < 0.001 for 60 and 120 min, and (*p* < 0.01 at 240 min).

### 3.3. Activation of Insulin Signaling Proteins in the Hypothalamus after an Acute Administration of DPIN

To address the action mechanism of DPIN in the cell, we first analyzed phosphorylation of the IRS-1, assuming that DPIN could be acting through the insulin receptor as an insulin sensitizer. As shown in Figure 4A, while there is no change in IRS-1 tyrosine 896, phosphorylation on serine 612 residue is significantly increased with DPIN administration (*p* < 0.05). Therefore, DPIN (500 mg/Kg) resulted in a clear inhibition of IRS-1, subsequently the balance among activating phosphorylation at tyrosine residues was decreased when compared with the inhibitory phosphorylation at serine residues. Next, we examined the PI3K. Here, we evaluated the phosphorylation of the regulatory subunit p85, necessary for the binding of the p110 catalytic unit to the membrane and consequent activation of PI3K. Western blot analysis of PI3K showed progressive activating phosphorylation of p85 at Tyr607. The PI3K activating phosphorylation increased significantly (*p* < 0.05) after 120 and 240 min of DPIN administration (Figure 4B). Afterward, we analyzed the phosphorylation stage of Akt, as being PI3K the major mode of Akt activation. The activating phosphorylation was evaluated through the Ser473 antibody. It showed a significant (*p* < 0.05) increase at the three evaluated points after DPIN administration, even when the total protein decreased significantly (*p* < 0.05) (Figure 4C).

According with an increased activity of the Akt, the phosphorylation at Ser9 of its substrate GSK-3β was also raised (*p* < 0.05) after DPIN intake, therefore resulting in the inactivation of GSK-3β (Figure 5A). The ratio between Ser9 and Tyr896 phosphorylation confirmed the inactivation of the kinase at 120 and 240 min (*p* < 0.05). When inactive, GSK-3β does not phosphorylate and thus activates the enzyme GS. However, our analyses show high and significant (*p* < 0.05) phosphorylation, and thereupon, inactivation of the GS after DPIN administration (Figure 5B). Moreover, the phosphorylation status at Ser2448 of the mTOR protein, a downstream substrate of Akt, was significantly (*p* < 0.05) increase after DPIN intake (Figure 4C), resulting in mTOR activation. By last, and because the phosphatases involved in regulating Akt activity (PTEN and PP2C) do not change in response to DPIN administration (Figure 5D), Akt activation via PI3K was confirmed.

### 3.4. Hypothalamic Glucose-Sensing Proteins Are Affected by Acute D-PIN

We evaluated the 3′,5′- cyclic adenosine monophosphate (cAMP)-dependent protein kinase A (PKA) as an AMPK upstream protein related to energy-sensing. We studied the threonine 198 phosphorylation of the PKA alpha catalytic subunit (PKSα), a necessary post-translational modification for PKA activation, which is also cAMP dependent. Our analysis (Figure 6A) does not show changes in terms of threonine 198 phosphorylation after DPIN administration. However, there is a drop of the PKA catalytic subunit transcription at 60 min (*p* < 0.05). We next examined the phosphorylation of AMPKα at Thr172 (Figure 6B), required for the protein activation. There are significant changes in terms of activating phosphorylation at 60 and 120 min (*p* < 0.05) post-administration, while there are no changes in total AMPKα quantity.

## 4. Discussion

Natural products with active ingredients that modulate glucose metabolism are gaining relevance in the management of chronic disorders that course with insulin resistance, including obesity-associated Type 2 Diabetes. Moreover, they have also been proposed for improving insulin resistance in chronic neurodegenerative disorders [32]. Insulin sensitizers such as inositols have been suggested for these purposes, although no studies addressed the impact of these compounds on insulin signaling in the hypothalamus, the main brain center for metabolic integration. As previously described in liver, DPIN played a positive role in the regulation of insulin-mediated glucose uptake in the liver through the translocation and activation of the PI3K/Akt signaling pathway in Type 2 Diabetes rats [12]. Our results with the natural carob fruit-derived inositol DPIN support this view. Acute DPIN administration resulted in the activation of the PI3K/Akt signaling pathway leading to improved insulin signaling. This action is produced downstream of the insulin receptor, since Tyr phosphorylation of IRS-1, one of its main substrates of the insulin receptor, was not phosphorylated after DPIN administration. Furthermore, the timing of DPIN’s activity on the PI3K/Akt signaling pathway corresponds to the plasma levels of the drug after oral delivery.

PI3K phosphorylation analysis shows that there is increased activation of the protein after DPIN administration. This inositol also increases the amount of activated Akt, and this phospho-Akt (at Ser473) inhibited the GSK-3β kinase activity directly. Analysis of GSK-3β kinase substrates was then addressed by examining GS phosphorylation. The inhibitory phosphorylation of GSK-3β found did not match with the phosphorylation of GS, which we expected to bereduced. In thisregard, we propose that the parallel increase of glucagon might be responsible for GS phosphorylation observed. Overall, after DPIN administration there is an activation in the hypothalamus of the kinase PI3K, independent of IRS-1 activation, followed by a clear engagement of the PI3K/Akt pathway.

Interestingly, we found that there is inhibitory phosphorylation of IRS-1 on Ser612. The phosphorylation of IRS-1 on Ser612 negatively regulates the function of IRS-1 and it is produced by the kinase activity of mTOR [33]. A plausible explanation is that the enhanced mTOR activity might be responsible for phosphorylation of IRS-1 on Ser612, acting as a negative feedback inhibition of the insulin receptor substrate, indicating a blockade of the insulin receptor pathway. The decreased circulating levels of insulin and the lack of changes on IGF-1, proposed as an Akt activator, indicate that the presence of DPIN substitutes insulin-growth factor signaling in the hypothalamus, a fact extremely interesting for conditions where insulin resistance in this metabolic integrator disrupts energy expenditure.

Likewise, we observe a significant decrease in insulin levels at 60 min of DPIN administration that recovery at 240 min. The increase found of DPIN is associated with a decrease in circulating insulin concentrations. That fact suggesting the DPIN is substituting insulin, inducing a physiological reduction of it demand from peripheral and central tissues. Since we did not observe changes in glycaemia, we examined the counter-regulatory response to hypoglycemia which is mediated, in part, by glucagon from the pancreatic islets [34]. We found a tendency to increase in glucagon after the acute administration of DPIN, causing in parallel a decrease in HOMA-IR and IGR values. All this while maintaining invariable the IGF-1 levels for all time points, thus this system would not be having influences on IRS-1. It is known that HOMA-IR is a validated index to evaluate the sensitivity to insulin [35] while IGR is proposed to predict the appropriate choice of glucose-lowering drugs for managing diabetes [36]. That is, treatments capable of increasing the IGR may be beneficial under insulinopenic conditions, while those that lower values of the IGR may be beneficial in conditions of hyperinsulinemia or insulin resistance. Our results of these validated indices support the proposal of a crucial PI3K/Akt activation signaling pathway in the hypothalamus by DPIN.

The net effect is a reduction of the insulin demand, but without leading neither to hypo- nor hyperglycemia, again indicating a beneficial effect by reducing the overload of the endocrine pancreas. These effects are associated with the activation of mTOR and AMPK in the hypothalamus, metabolic sensors modulating energy expenditure, and that increase appetite [37,38]. Activated AMPK inhibits ATP consumption and stimulates ATP generation [39]. Hormones and pharmacological agents can also modulate AMPK activity in the hypothalamus. In this sense, previous work [40,41,42] confirmed the role of ghrelin as one of the signals of energy deficit that enhance hypothalamic AMPK activity to increase feeding. Consistent with our results we have recently described that DPIN activates ghrelin release to control endocrine pancreas activity [26] consequently activating mTOR and AMPK, both stimulators of body weight gain. Further studies using a nutritional supplement with DPIN might be needed to identify whether this inositol might generate weight gain under insulin resistance and/or food deprivation conditions.

In addition to the effects on the PI3K/Akt pathway, inositols or inositol-glycans have been demonstrated to regulate phosphatases that modulate their action as insulin mimetics. Some studies link phosphatases with the action of inositol glycan containing DPIN. Moreover, Akt activation is tightly controlled by the counteraction of phosphatases. To examine some of these phosphatases we selected phosphatase and tensin homolog (PTEN), a phosphatase that can shut down PI3K/Akt pathway [43], and protein phosphatase 2C (PP2C), which is capable of modulating the actions of inositol-derived glycans on insulin secretion [44] through is actions as an allosteric modulator of PP2C [45]. We did not find a modulation PTEN phosphatase protein expression after DPIN administration, and we only detected a small decrease of total PP2C protein in the hypothalamus 240 min after DPIN administration, suggesting that if any, the effect might be inhibitory. Since we could not measure specific phosphatase activity we cannot discard an allosteric activation/inhibition of these phosphatases, so the potential participation of either PTEN or PP2C in the regulation of PI3K/Akt activation remains partially unsolved.

## 5. Conclusions

Nowadays, the complex relationship between insulin, glucose metabolism and inositol-related molecules is poorly understood. Our work aimed to approach this convoluted network studying the activation of the PI3K/Akt signaling pathway in the hypothalamus as insulin improving hub. In summary, the present study reveals that DPIN is an orally active inositol that has an activatory effect on insulin signaling in the medial basal hypothalamus being independent of the insulin secretion and/or its receptor. DPIN triggers this cell pathway causing activation of Akt and in turn, the substrates of this kinase: GSK-3β, mTOR, and GS. By last, brain insulin signaling is known to decrease with aging. Insulin resistance has been identified as one of the major risk factors in neurodegenerative diseases where it has recently been linked to the hyperphosphorylation of the protein tau, a hallmark for Alzheimer’s Disease. This suggests that improving insulin sensitivity might be beneficial to patients with brain insulin-resistance associated disorders, such as Alzheimer’s disease, an hypothesis that deserves further experimental studies.

## Figures and Tables

**Figure 1 nutrients-13-02268-f001:**
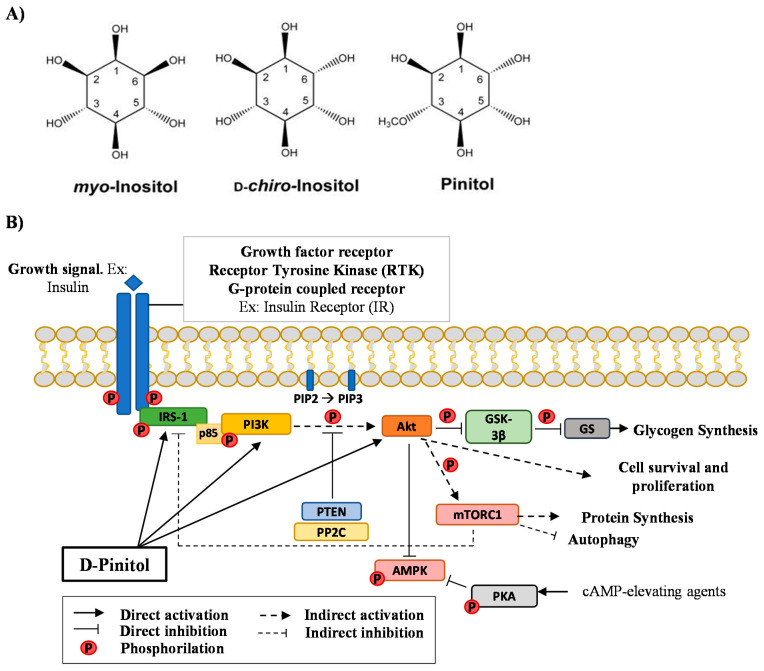
(**A**), Chemist structure of *myo*-Inositol, D-*chiro*-Inositol, and Pinitol (DPIN). DPIN is the 3-*O*-methyl form of D-chiro-inositol (DCI) and is found as a cyclitol, a cyclic polyol. (**B**), Schematic diagram depicting the signaling of the Phosphatidylinositol-3-kinase/Protein kinase B (PI3K/Akt) pathway via Insulin Receptor Substrate 1 (IRS-1). Glycogen synthase kinase 3β (GSK-3β); glycogen synthase (GS); Tensin homolog (PTEN); Protein phosphatase 2C (PP2C); Adenosine monophosphate-activated protein kinase (AMPK); protein kinase A (PKA) and mammalian Target of Rapamycin Complex 1 (mTORC1).

**Figure 2 nutrients-13-02268-f002:**
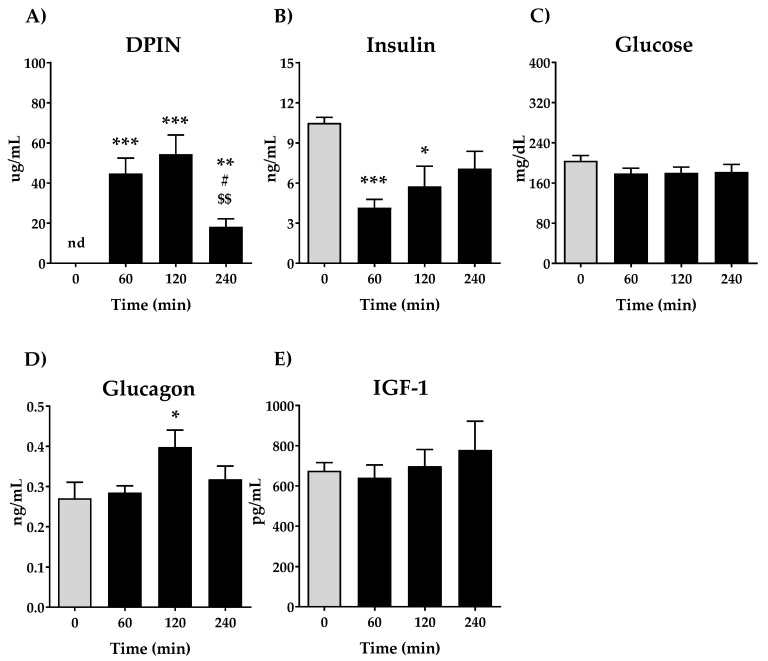
Effects of acute oral administration of DPIN (500 mg/Kg) on plasma levels of; (**A**), DPIN, nd = no detected; (**B**), Insulin; (**C**) Glucose; (**D**), Glucagon and; (**E**), Plasma Insulin-like growth factor-1 (IGF-1) on Wistar rats. Histograms represent the mean ± standard error of the mean (SEM) (*n* = 8). One-way ANOVA and Tukey’s test: (*) *p* < 0.05, (**) *p* < 0.01, (***) *p* < 0.001 vs. 0 min control group; (#) *p* < 0.05 vs. 60 min group. ($$) *p* < 0.01 vs. 120 min group.

**Figure 3 nutrients-13-02268-f003:**
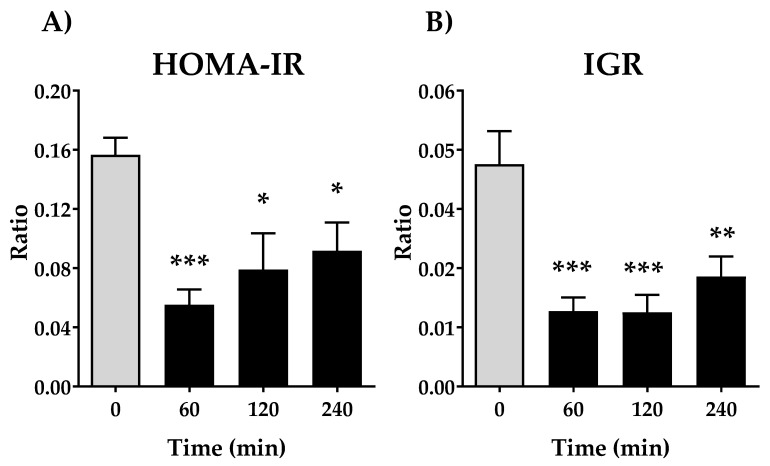
Effects of acute oral administration of DPIN (500 mg/Kg) on Homeostatic model assessment for Insulin Resistance (HOMA-IR) (**A**), and insulin-glucagon ratio (IGR) (**B**), on Wistar rats. Histograms represent the mean ± SEM (*n* = 8). One-way ANOVA and Tukey’s test; (*) *p* < 0.05, (**) *p* < 0.01, (***) *p* < 0.001 vs. 0 min control group.

**Figure 4 nutrients-13-02268-f004:**
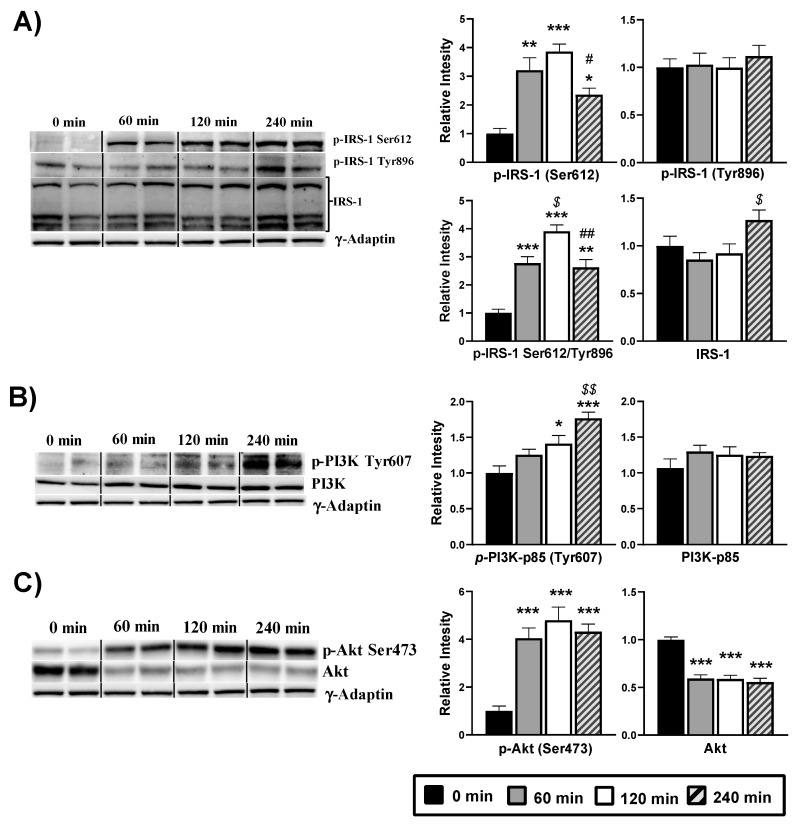
Western blot analysis of the phosphorylation status of the insulin signaling proteins from hypothalamus lysates of Wistar rats treated with 500 mg/Kg of D-Pinitol (DPIN) for 0, 60, 120 and 240 min. Histograms show the effect of this acute administration on; (**A**), Insulin receptor substrate 1 (IRS-1) phosphorylation on serine 612, tyrosine 896, p(Ser)-IRS-1/p(Tyr)-IRS-1 ratio, and quantity of total IRS-1; (**B**), p85 regulatory domain of the phosphatidylinositol 3 kinase (p85-PI3K) phosphorylation on tyrosine 607, and quantity of total p85-PI3K; (**C**), Protein Kinase B (Akt) phosphorylation on serine 473, and quantity of total Akt (Appendix A for additional information). The corresponding expression of γ-Adaptin is shown as a loading control per lane. All samples were obtained at the same time and processed in parallel. Adjusting to digital images did not alter the information contained therein. Histograms represent the mean ± SEM (*n* = 8) and they have their respective Western blot membranes next to. One-way ANOVA and Tukey’s test was performed: (*) *p* < 0.05, (**) *p* < 0.01, (***) *p* < 0.001 vs. 0 min control group; ($) *p* < 0.05, ($$) *p* < 0.01 vs. 60 min group; (#) *p* < 0.05, (##) *p* < 0.01 vs. 120 min group.

**Figure 5 nutrients-13-02268-f005:**
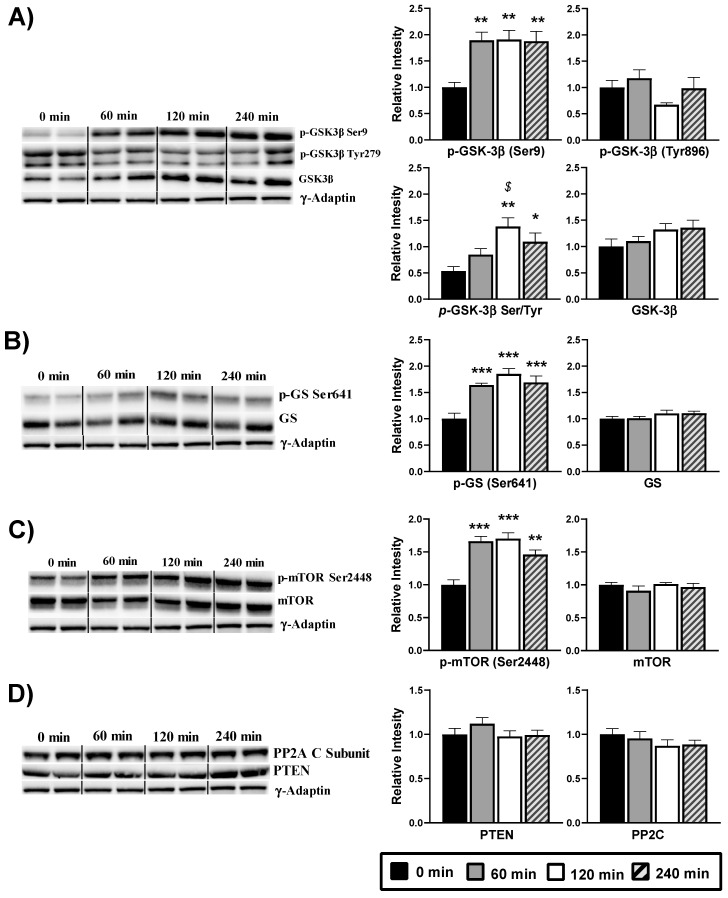
Western blot analysis of the phosphorylation status of the insulin signaling proteins from hypothalamus lysates of Wistar rats treated with 500 mg/Kg of D-Pinitol (DPIN) for 0, 60, 120 and 240 min. Histograms show the effect of this acute administration on; (**A**), Glycogen synthase kinase 3β (GSK-3β) phosphorylation on serine 9, tyrosine 896, p(Ser)- GSK-3β /p(Tyr)- GSK-3β ratio, and quantity of total GSK-3β; (**B**), Glycogen synthase (GS) phosphorylation on serine 641, and quantity of total GS; (**C**), mammalian Target of Rapamycin (mTOR) phosphorylation on serine 2448, and quantity of total mTOR; (**D**), phosphatases involved in regulating Akt activity (PTEN and PP2C). The blots shown are a representation of all the bands. (Appendix A for additional information). The corresponding expression of γ-Adaptin is shown as a loading control per lane. All samples were obtained at the same time and processed in parallel. Adjusting to digital images did not alter the information contained therein. Histograms represent the mean ± SEM (*n* = 8) and they have their respective Western blot membranes next to. One-way ANOVA and Tukey’s test was performed: (*) *p* < 0.05, (**) *p* < 0.01, (***) *p* < 0.001 vs. 0 min control group; ($) *p* < 0.05 vs. 60 min group.

**Figure 6 nutrients-13-02268-f006:**
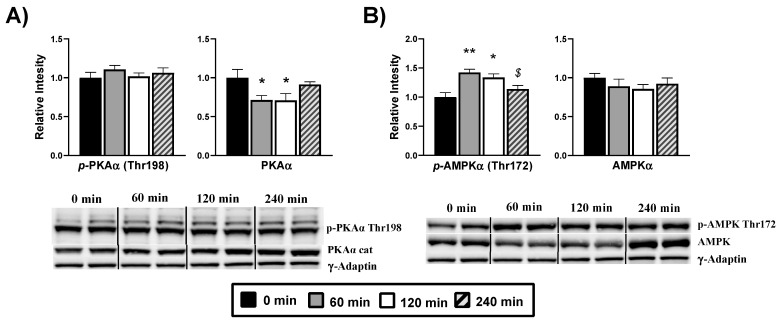
Hypothalamic glucose-sensing proteins are affected by acute D-Pinitol (DPIN) administration (500 mg/Kg). Samples were collected at 0 (controls), 60, 120, and 240 min after the administration of DPIN. Histograms show the effect of this acute administration on; (**A**), Alpha protein kinase A (PKAα cat) catalytic subunit phosphorylation on threonine 198 and total quantity of PKAα; (**B**), Adenosine monophosphate-activated protein kinase (AMPK) phosphorylation on threonine 172 and total quantity of AMPK. The blots shown are a representation of all the bands (Appendix A for additional information). The corresponding expression of γ-Adaptin is shown as a loading control per lane. All samples were obtained at the same time and processed in parallel. Adjusting to digital images did not alter the information contained therein. Histograms represent the mean ± SEM (*n* = 8) and they have their respective Western blot membranes next to. One-way ANOVA and Tukey’s test was performed: (*) *p* < 0.05 and (**) *p* < 0.01 vs. 0 min control group; ($) *p* < 0.05, vs. 60 min group.

## Data Availability

The data presented in this study are available on request from the corresponding author.

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
