# Peer review of "Activation of PI3K/Akt Signaling Pathway in Rat Hypothalamus Induced by an Acute Oral Administration of D-Pinitol"

_nutrients, 2021, doi:10.3390/nu13072268_

Round 1

Reviewer 1 Report

The manuscript considers the mechanisms involved in hypothalamic insulin signaling pathway after oral D-pinitol administration in rats. Although the impact of inositols on insulin signalling has been already documented in literature, the knowledge was lacking regarding effects on hypothalamus, which acts as the main brain centre for metabolic integration. The present investigation, by means of evaluations of specific post-receptor downstream pathways, demonstrates that the mechanism of action of D-pinitol at hypothalamic level is linked to the PI3K/Akt signaling pathway, which suggests that hypothalamus has a role as insulin improving hub, strictly paralleled to plasma concentrations of D-pinitol. Moreover, since insulin resistance has been linked to neurodegenerative diseases, the present study provides mechanistic justification for a treatment of brain insulin-resistance associated disorders.

The study has been carefully conducted, using a rat model, following European Directives on animal protection. Methods are exhaustively stated, both regarding procedures ad variables evaluated; the statistical analysis appears adequately performed, although some aspects could be further addressed (see below). Results are clearly presented and discussed.

The statistical issues that could be evaluated:

Figures 2 & 3: the statistical analysis was obtained with ANOVA followed by Bonferroni’s post hoc test. Why was this test used, instead of Tukey’s HSD test, also mentioned in methods? Figures 4-5-6 report the Tukey’s test, which is a complete post hoc test. Bonferroni test is used for planned comparisons and tends to be overly conservative.

lines 259-275: the significance level was always set as p<0.05, although in figure 4 the significance levels are differentiated (p<0.05, p<0.01, p<0.001). Is this a way to simplify the interpretation of the results?

lines 290-300 and 317-326-: same comment as above.

Figure 2A: The horizontal line over the histograms relative to 60 and 120 minutes is not clear; the asterisks (***), according to the legend, indicate significance vs 0 min control groups, but in this way, it seems a comparison between 60 and 120 min.  It would be clearer if asterisks were shown over each histogram. Same comment for fig. 3, fig. 4, fig. 5 and fig. 6. Moreover, in fig. 4, fig. 5 and fig. 6 legends, the “p” should be italicized, as well as at lines 323 and 325.

Some other minor points:

line 204: were analyzed by an autoanalyzer

line 214: insulin-glucagon ratio

line 218: One-way analysis of variance

line 242: Bonferroni’s

Author Response

Dear reviewer

Below I respond to your comments to the manuscript:

Point 1 reply: All figures are now run with Tukey’s HSD test as its post hoc test. Figure footnotes and statistics of Figure 2&3 have been modified in accordance.

Point 2 reply: Yes, p<0.05 is a way to simplify the interpretation of the results.

Point 3 reply: We appreciate referee’s suggestion, we have changed it and now the asterisks are shown over each histogram.

Point 4 reply: We appreciate referee’s suggestion, we have reviewed it.

Point 5 reply: We appreciate referee’s suggestions, we have reviewed it.

All changes have also been marked in the new manuscript.

Thank you for your time

Best regard

Dr. Juan Decara

Reviewer 2 Report

The authors have studied the D-pinitol-induced activation of PI3K/Akt signaling pathway in the rat hypothalamus and have concluded that D-pinitol has an activatory effect on insulin signaling in the medial basal hypothalamus independent of the insulin secretion and/or its receptor. The paper is well-written.

  1. Can the authors suggest, based on their findings, some biomarkers with predictive value for brain insulin resistance in cognitive decline and (or) neurodegenerative disorders?
  2. line 428: correct nd/or to and/or.
  3. Check that the references are correctly written and that page numbers/article IDs are mentioned. Either provide DOIs for all references or delete the DOIs (the style required by the journal is American Chemical Society).

Author Response

Dear reviewer

Below I respond to your comments to the manuscript:

Comments and Suggestions for Authors 1 reply: We appreciate referee’s suggestion, we have reviewed it. Please see in yellow the changes in the conclusion section.

Comments and Suggestions for Authors 2 reply: We appreciate referee’s suggestion, we have corrected it.

Comments and Suggestions for Authors 3: We appreciate referee’s suggestion, we have reviewed it. Please see in yellow the changes.

All changes have also been marked in the new manuscript.

Thank you for your time

Best regard

Dr. Juan Decara

Reviewer 3 Report

This well conducted study investigated the effects of orally administered D-Pinitol on hypothalamic insulin signalling. The changes in the insulin signalling mechanism in the hypothalamus are correlated with changes in circulating insulin and glucagon. However, insulin signalling is the same in all insulin sensitive tissues so the changes in circulating insulin and glucagon could have been the result of D-Pinitol acting peripherally.

Leptin also signals through some of the same signalling molecules as insulin including the PI3K-AKT, AMPK and mTOR pathways (Kwon O et al leptin signalling pathways in hypothalamic neurons . Cell Mol Life Sci 73:1457-77  2016. Both insulin and Leptin inhibit food intake. This should be mentioned in the discussion section that discuses food intake.

Minor points

  1. There are multiple places where the manuscript would be improved by review of a native English speaker.
  2. The legend to figure 1 is too long. Better to just say what the figure shows in the legend and then describe the signalling pathway in the text.
  3. The statement “DPIN played a positive role in the regulation of insulin-mediated glucose uptake in the liver “ is incorrect. There is no insulin mediated glucose uptake in the liver. Insulin signals in the liver but the action if to inhibit gluconeogenesis and glycogenolysis. GLUT 4 the insulin sensitive glucose transported is not expressed in the liver.
  4. In section 2.2 the administration of D-Pinotol to rats could be described more clearly
  5. In Figure 4A IRS1 appears to be in the wrong location please check.

Author Response

Dear reviewer

Below I respond to your comments to the manuscript:

Comments and Suggestions for Authors 1 reply: We appreciate the referee’s comments. Whether the activation of the IRS-PI3K-AKT-GSK3 cascade on the hypothalamus that we are describing here are derived from direct actions of D-Pinitol in relevant hypothalamic nuclei, or arise from peripheral alterations on hormones targeting the hypothalamus, cannot be absolutely concluded from the present study. D-Pinitol action is indeed mediated by multiple mechanisms, as described in Navarro et al. (Nutrients, 2020) that include hormonal alterations. However, our findings cannot attribute to insulin the activatory phosphorylation or IRS1, nor any of the described alterations in hormone secretion: insulin is reduced, and IGF-1 and leptin are not altered. Although Glucagon and Ghrelin are activated, some other peripheral hormones and signals eventually might activate this signaling cascade. However, the scientific literature has clearly established an insulin mimetic action for D-Pinitol, so our working hypothesis is that is this polyol the main responsible actor for the insulin receptor-independent activator of the IRS-PI3K-AKT-GSK3 cascade

Comments and Suggestions for Authors 2 reply: We appreciate the referee’s suggestion, we have added it (reference 27).

Minor Point 1 reply: We appreciate the referee’s suggestion, we have reviewed it.

Minor Point 2 reply: We appreciate the referee’s suggestion, we have changed it as suggested.

Minor Point 3 reply: We appreciate the referee’s recommendation, in fact, the statement is wrong due that the cited publication was not correct. There was a little confusion that has been corrected. You can see modified reference marked in yellow in the text.

Minor Point 4 reply: We appreciate the referee’s suggestion, we have reviewed it. Please see the changes in 2.2 section

Minor Point 5 reply: We appreciate the referee’s suggestion, we have corrected it.

All changes have also been marked in the new manuscript.

Thank you for your time

Best regard

Dr. Juan Decara